# High-performance liquid chromatographic method for determination of voriconazole in pure and gel formulation

Samina Sheikh[1], Anwar Ejaz Beg[2], Mirza Tasawer Baig[1], Sadaf Ibrahim[3], Ambreen Huma[4], Aisha Jabeen[3], Zubair Anwar[5]*

1 Department of Pharmacy Practice, Faculty of Pharmacy, Ziauddin University, Karachi, Pakistan,
2 Department of Pharmaceutics, Faculty of Pharmacy, Ziauddin University, Karachi, Pakistan, 3 Department of Pharmacology, Faculty of Pharmacy, Ziauddin University, Karachi, Pakistan, 4 Department of Pharmacognosy, Faculty of Pharmacy, Ziauddin University, Karachi, Pakistan, 5 Department of Pharmaceutical Chemistry, Baqai Institute of Pharmaceutical Sciences, Baqai Medical University, Gadap Town, Karachi, Pakistan

* zubair_ana@yahoo.com, Zubair_ana@baqai.edu.pk

**Data Availability Statement:** All relevant data are within the manuscript.

## Abstract

For routine measurement of Voriconazole (VZ) in a pure and gel formulation, a quick and accurate RP-HPLC technique with UV detection (254 nm) was developed. With a flow rate of 1.0 ml/min using a mobile phase that contained acetonitrile and water mixed 50:50, v/v. Internal standard approach was used for quantification. The method shows good linearity (correlation coefficient = 0.9999) with acceptable accuracy, precision and robustness. Three elements were taken into account to measure robustness. Flow rate, mobile phase composition, and pH all have an impact on the response, but only the flow rate which causes a reduction in the concentration of the drug—has a significant impact on the response. Analyst, equipment, and days were taken into consideration for a precision measurement. The analytical procedure had good precision, as seen by the %RSD which is found to be less than 2.0. The proposed method was straightforward, extremely sensitive, exact, and accurate, and it had a retention time of less than 4 minutes, indicating that it is appropriate for daily quality control.

## Introduction

Voriconazole (VZ) (Fig 1), is an extensive-spectrum antifungal drug from the triazole class [1], against a variety of fungal infections, such as those caused by Aspergillus and Candida species [2]. VZ, which is structurally derived from fluconazole [3] (Fig 1), is used for a crucial treatment of invasive fungal infections, especially in individuals with impaired immune systems, due to its increased potency and wider range of action [4]. VZ is a white, crystalline powder that dissolves very well in organic solvents like acetone and methanol but weakly in water [5]. Its distinguishing chemical characteristic is a triazole ring that binds tightly to fungal cytochrome P450 enzymes and inhibits the synthesis of ergosterol, an essential component of fungal cell membranes [6].

**Funding:** The author(s) received no specific funding for this work.

**Competing interests:** The authors have declared that no competing interests exist.

**Abbreviations:** VZ, Voriconazole; HPLC, High Pressure Liquid Chromatography; LOD, Limit of detection; LOQ, limit of quantification; FDA, Food and Drug Administration; TEA, Tri-ethanol amine.

VZ has a strong therapeutic profile that works against a variety of fungal infections, therefore it can be used in various dose formulations [7]. It is currently offered in several forms, including suspensions, intravenous injections, and oral tablets [8]. The analytical evaluation of VZ is crucial for assuring its effectiveness and quality in pharmaceutical formulations. Various analytical techniques have been used to estimate VZ in its pure and in pharmaceutical formulations, these techniques include high-performance liquid chromatography (HPLC) [9–11], ultra-performance liquid chromatography (UPLC) [12–20], Liquid chromatography-tandem mass spectrometry(LC-MS/MS) [21–30], high-performance tin-layer chromatography (HPTLC) [10, 31–38], UV-visible spectroscopy [39–48] and capillary electrophoresis [9, 49–56]. HPLC is the most extensively employed because of its sensitivity, specificity, and reproducibility [57–63]. HPLC techniques commonly use reverse-phase columns, such as C18 with a UV detector to detect VZ [10]. It is essential to optimize variables including the composition of the mobile phase, flow rate, and detection wavelength to precisely quantify and separate VZ from any contaminants or degradation products [11].

Each technique has distinctive benefits and precise applications, depending on the desired level of sensitivity, sample complexity, and available instrumentation. Because it strikes an acceptable balance between accuracy and effectiveness, HPLC is still the industry standard, especially for routine quality control [64]. The details of the previously developed HPLC methods for the determination of VZ in pure and pharmaceutical formulations are given in Table 1. Previous HPLC methods developed for the determination of VZ are time-consuming, tedious, expensive and need pretreatment of the samples. So, therefore, the present study aims to develop and validate an easy, affordable, sensitive, reliable, precise and robust HPLC method for the determination of VZ in pure and gel formulations.

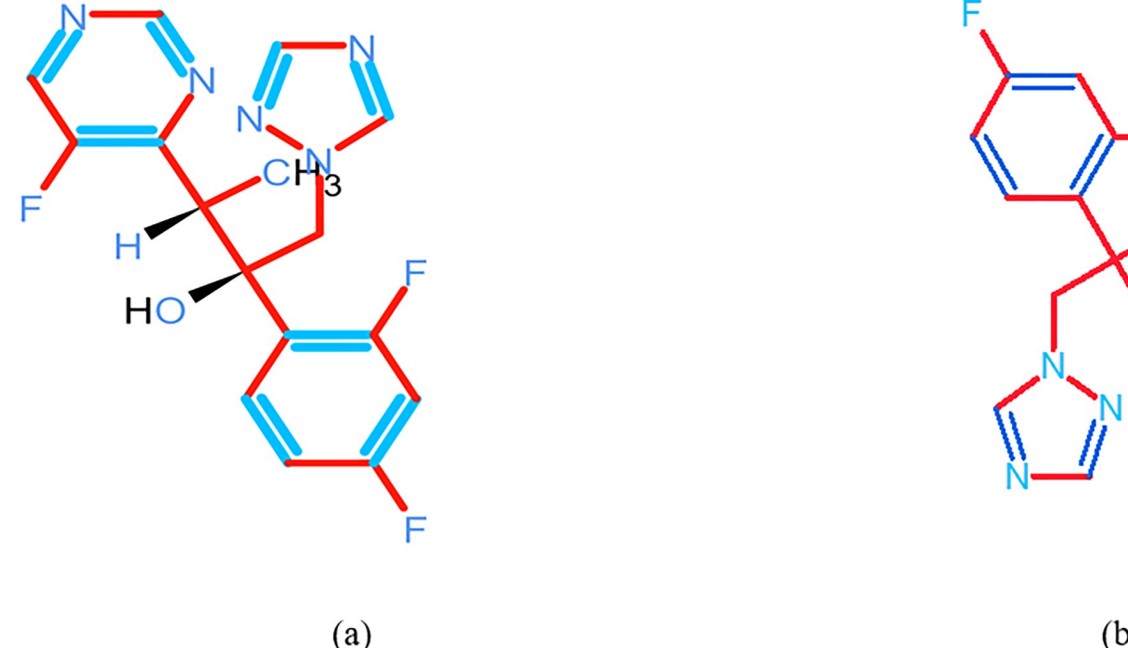

(a)                                                                                  (b)

**Fig 1.** Chemical structure of voriconazole (a) and fluconazole (b).

**Table 1. Details of the previously reported HPLC method for the determination of VZ.**

| Solvent System | Column | Flow Rate (ml/min) | tR (min) | Detection Wavelength (nm) | Range | LOD | LOQ | Corr. Coeff. | Accuracy (%) /Precision (% RSD) | References |
|---|---|---|---|---|---|---|---|---|---|---|
| Acetonitrile and ultrapure water (50:50) (v/v) | C18 column | 1.0 | 4.1 | 256 | 1–30 µg/ml | 0.022 µg/ml | 0.065 µg/ml | 0.999 | 99.49 and 100.7% | [65] |
| Isopropyl alcohol: water (80:20) v/v | C18 column | 1.0 | 7.92 | 256 | 5–25 µg/ml | 0.57 | 1.54 | **0.999** | 99.102 to 99.724% | [66] |
| Acetonitrile and water in the ratio of 60:40V/V. | C18 column | 1.0 | 5.360 | 256 | 10–50 µg/mL | 0.749 | 2.24 | 0.999 | 99.89–100.86% (<2%) | [67] |
| Acetonitrile: ultrapure water (60:40, v/v) | C18 column | 0.8 | 3.20 | 255 | 0.125–16.0 µg/mL | 0.125 µg/mL | 0.25 µg/mL | 0.9999 | 5.19 to 8.96% and from −13.12% to 8.04% | [68] |
| 0.05 mol L-1 disodium hydrogen phosphate buffer): acetonitrile (1:1, v/v) | C18 column | 1.0 | - | 255 | 6.0–60 µg mL-1 | 2.11 | 7.25 | 0.999 | 100.7% | [69] |
| Acetonitrile and water (50:50, v/v) | C18 column | 1.0 | ≤ 4 | 260 | | - | 0.55 | 2 | 0.9999 | 99.45 and 100.50%, mean R. S.D.% = 0.53%, | [70] |
| Ammonium phosphate dibasic buffer–acetonitrile (52:48, v/v) | C18 column | 1.0 | - | 250 | | - | - | - | 0.999 | 98.8% to 100.4% R.S.D = 0.18% | [71] |
| Acetonitrile with filtered 0.04 M ammonium dihydrogen phosphate buffer | C18 column | 0.8 | 7.5 | 255 | 0.2 mg/ml | - | - | - | - | [72] |
| Acetonitrile.water. acetic acid (55:45:0.25, v/v/v) | C18 column | 1.0 | - | 256 | - | - | 0.10 µg/mL | - | 8.6% and 6.0% | [73] |
| Acetonitrile and water (7:3) | C18 column | 1.0 | - | 255 | 0.2–15 mg/L (0.57–43.0 µmol/L) | - | 0.2 mg/L (0.57µmol/L) | 0.998 | 80–120% | [74] |
| Methanol: triethylamine solutions 0.6%, (50:50, v/v) | 100 RP-8 column | 1.0 | 3.0 | 255 | 20.0–100.0 µg mL−1 | - | - | 0.9999 | 100.4% R.S.D = ≤ 1.0% | [75] |
| Acetonitrile at a ratio of 40 to 60 (v/v) | LiChrospher-100 RP-18 | 1.0 | 3.2 | 254 | - | - | - | - | - | [76] |
| Acetonitrile and acetic acid solution (50:50 v/v) | C18 column | 1.0 | - | 256 | 0.1 and 50 µg/mL | 0.031 µg/mL | 0.093 µg/mL | - | 98.99–102.34% | [64] |
| 0.05 M ammonium acetate/acetonitrile/methanol 40:20:40 (v/v/v) | MV C18 column | 1.0 | 7.45 | 256 | 0.1–10 mg/L | ~0.06 µg/mL | - | - | 97 to 106% | [77] |
| Acetonitrile, methanol, and 0.1% aqueous trifluoro acetate buffer 15:30:55 (v/v/v) | C18 column | 1.0 | 5.30 | 256 | 0.25281–1.51690 µg mL−1 | - | - | - | 89.3 to 100.3% | [78] |
| Acetonitrile: water (40:60, v/v) | Hypersil C18 column | 1.0 | - | - | 5–25 µg/m | 0.1841 µg/ml | 0.5581 µg/m | 0.9923 | 99.74–100.39 R.S.D = < 2% | [79] |

## Materials and methods

### Chemicals

VZ (>98.0%) was purchased from Sigma Aldrich (St. louis, MO, USA). All the solvents and reagents used in this study were of HPLC grade and used without any prior purification.

### pH measurements

pH measurements of VZ were carried out by LCD pH meter, (Elmetron CP-500, sensitivity ±0.01 pH units Poland). Calibration of pH meter was carried out by using commercially available buffer tablets of pH 4.0, 7.0, and 9.0.

### Preparation of gel formulation

VZ dissolved in ethanol with the help of a magnetic stirrer for 30 minutes until a clear solution was formed. Different excipients were used to prepare the hydrogels at different concentrations that is given in Table 2. Every time a precisely measure quantity of carbomer was added in small amounts to a glass container filled with water, stirring continuously to prevent the formation of lumps using a glass rod and a mechanical mixer for 15 min at a speed of 500 rpm to completely dissolve the polymer this would result in the formation of a gel. A solubilizing and pH-adjusting triethanol amine (TEA) is added until the desired pH is obtained, and the remaining water and humectant are added and mixed thoroughly for a further 30 min. VZ solution was added at the end of the stirring for a further 10 min. The formulated gel is allowed to settle for 24 hours before subjecting them to analysis. For each formulation, the placebo gel was likewise made identically [80, 81].

### Chromatographic condition

The HPLC apparatus was Shimadzu chromatographic system (20A autosampler) with a variable wavelength programmable UV-Vis detector, a reversed-phase column (L1, 5μ) 4.6× 150mm. Prepare a mixture of acetonitrile and water (1:1, v/v). Stirrer for 5 minutes and allow to equilibrate at room temperature. Filter this solution through a 0.22-micron membrane filter and use the filtrate as a mobile phase. Acetonitrile and water (50:50 v/v) served as the mobile phase, and the flow rate was 1 mL per minute. The temperature was set to 30±1˚C, and the UV detector's wavelength was 254 nm.

### Preparation of reference standard solution

Weight accurately 0.05 mg (50 μg) of the reference standard of VZ into 100 mL of volumetric flask. Add 100 ml of mobile phase and dissolve it by stirring.

Table 2. Composition of the formulated gel.

| Material Name | g/100gm |
|---|---|
| Voriconazole | 3.0gm |
| Propylene glycol | 15.000gm |
| Propylene glycol 400 | 10.000gm |
| Acrypol 940 (Carbomer) | 0.375gm |
| Ethanol | 30ml |
| Triethanolamine | 0.15gm |
| DI water | QS. 100gm |

## Preparation of test solutions

Weigh the gel about 2gm (equivalent to about 60mg of voriconazole) and transfer it accurately to a 200mL volumetric flask, adding 50mL of diluent dissolved by mechanical shaking. Make up the volume with diluent and mix. Filter a portion of this solution through a 0.22-micron membrane filter and use the filtrate as a test solution.

## Method validation

The proposed HPLC method was validated according to the guidelines of the International Council of Harmonization [82]. The details of the method validation parameter are discussed below:

**System suitability.**   Inject 5 replicates of the reference solution (45 μg/100 mL) and check the system suitability criteria for tailing factor and relative standard deviation. Inject in triplicate the test solution and record the chromatograms.

**Linearity and range.**   The plot of peak Area (AU) versus the corresponding concentrations of the prepared solutions (3.5–45 μg/mL) has been prepared. The prepared calibration curve is used to calculate the slope, intercept, standard deviation, standard deviation, standard error and correlation coefficient. The linearity of the calibration curves was used to determine the range for the assay of VZ.

**Accuracy.**   The accuracy of the proposed method was assessed by making known concentrations of VZ from the standard solutions (15.0, 30.0, and 45.0 μg/100 mL) from the stock solution. The known concentrations were made in duplicate and put through an assay to see the accuracy of the method.

**Precision.**   Dilutions of VZ (30.0μg/100 mL) were prepared to check the repeatability (intra-day) and intermediate (inter-day) precision of the method. To assess the precision of the suggested method, relative standard deviation (%RSD) was computed.

**Limit of detection (LOD) and limit of quantification (LOQ).**   The standard deviation of the slope was used to calculate the LOD and LOQ of the procedure by using the formulas:

$$LOD = 3 : 3 \times \sigma/S$$

$$LOQ = 10 \times \sigma/S$$

Where, σ = standard deviation of intercept and S = slope.

**Robustness.**   To test the robustness of the method, minor adjustments were made to the assay's parameters. By adjusting the flow rate (±0.2 mL/min), temperature (±5°C), and wavelength (±2 nm), the accuracy and precision of the procedure were estimated.

**Specificity.**   It measured the desired components without the interference of other species that might be present; separation is not necessarily required. Placebo, mobile phase and related substance solutions were prepared. No interfering peaks from the mobile phase and placebo were observed.

## Application of the proposed HPLC method

The developed HPLC method was applied to the prepared gel formulation of VZ containing (0.5, 1.5 and 3.0 gm). Each gel formulation was weighed and solubilized in ethanol and then afterwards subjected to HPLC analysis

## Result and discussion

### Method validation

The developed and validated HPLC method improves on previously reported HPLC methods by addressing the demands of sensitivity, selectivity, accuracy, precision, robustness, and specificity for the quantification of VZ. Various parameters have been studied to optimize the analysis conditions, including solvent system composition (acetonitrile:water, 10:90, 15:85, 20:80, 25:75, 30:70, 40:60, and 50:50, v/v), pH of the mobile phase (2.0, 2.5, 3.0, 3.5, 4.0, 4.5, 5.0, and 6.5), and flow rate of the mobile phase (0.1, 0.2, 0.3, 0.4, 0.6, 0.8, 1.0 mL/min). After multiple trials using the above-mentioned conditions, the proposed method has been validated using acetonitrile:water (50:50, v/v) with a flow rate of 1.0 mL/min at pH 5.0 to obtain accurate, precise, robust, reliable, and specific results for the estimation of VZ.

**System suitability.** Before the method validation, the system suitability of the proposed method has been determined and the results obtained are reported in Table 3. The results obtained are found to be in the acceptable range with a $t_R$ of 3.782 min (Fig 2). The %RSD was found to be less than 2% and the theoretical plates were found to be 2196 with a tailing factor of less than 1.110 (Table 3).

**Linearity and range.** The proposed method was found to be linear in the concentration range of 3.5–45.0 μg/100 mL. Table 4 presents the statistical analysis of these findings. With correlation coefficients ($R^2$) of 0.9999, respectively, there is a negligible scatter in the calibration curves' points (Fig 3). The purity of these compounds is shown by the fact that the y-intercepts for the measurement of VZ are near zero.

**Accuracy.** The proposed method's accuracy in determining VZ was 99.76±0.68, 100.0 ±0.37, and 100.1±0.18 respectively. The results show that the estimated concentrations and the applied concentrations agree, according to the results. In Table 5, findings from the method's accuracy assessment are shown. The relative accuracy error (%), standard deviation, and percent RSDs were found to be minimal, demonstrating the method's high degree of accuracy for identifying these compounds.

**Precision.** It measures the dispersion between chemical results made after many samples were taken with identical concentrations and under the same circumstances. The results of the precision studies are shown in Table 6, and the fact that the %RSDs was less than 1% showed that this technique is suitable for determining VZ.

**Limit of detection.** It is the minimum concentration of the analyte that could be identified but not quantified. The LODs for the assay of VZ were found to be 1.06 μg/100 mL, respectively (Table 4), this demonstrates that this approach is quite sensitive for identifying all of these compounds.

**Limit of quantification.** It involves quantifying the analyte's lowest concentration and drugs with good precision and accuracy. The values of LOQs for determining VZ were found to be 3.22μg/100 mL, respectively.

**Robustness.** It demonstrates that the testing method is unaffected by any little intentional alterations and demonstrates the method's compatibility and durability. The deliberate changes in temperature (±5˚C), wavelength (±2 nm) and flow rate (± 0.2 mL/min), as well as the

Table 3. System suitability parameters for the proposed HPLC method.

| | |
|---|---|
| **Retention time ($t_R$), min** | 3.782 |
| **%RSD** | 0.15 |
| **Theoretical plates** | 2196 |
| **Tailing factor** | 1.110 |

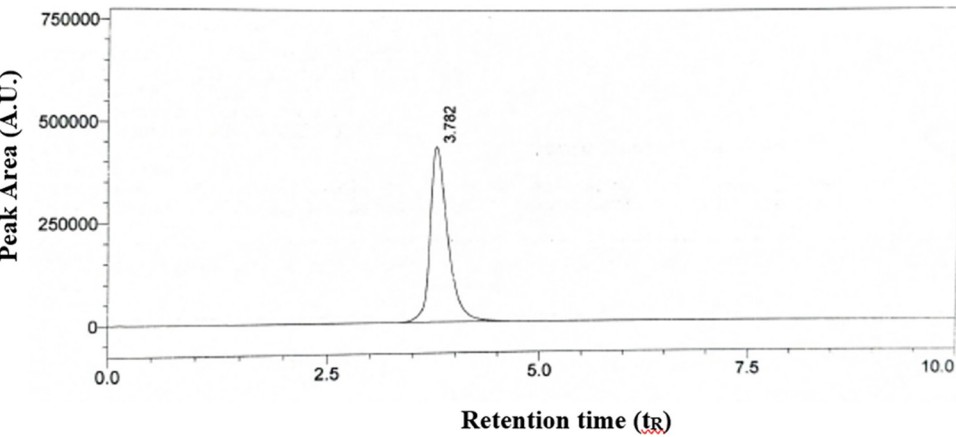

**Fig 2. Chromatogram of system suitability for the proposed HPLC method.**

accuracy and precision data collected following those adjustments, are given in Table 7. The accuracy and precision (%RSDs) for the identification of VZ after deliberate changes are 99.86–100.01,98.99–100.1, 99.99–100.1 and 0.11–0.22, 0.32–0.88, 0.44–0.51, respectively. The statistical comparison between the proposed method conditions and after making the deliberate changes has been carried out (Table 7). It has been found that there is no significant difference between the proposed method conditions and the deliberate changes made. The calculated $t$ values are less than that of tabulated $t$ values indicating that there is no significant difference and confirming that the method is reliable, sensitive, accurate, and precise. These outcomes demonstrate that the modifications made to the measurement have no impact on the method's accuracy and precision.

**Specificity.** The ability of the analytical method to detect and quantify the analyte of choice in the presence of impurities and degradation products is how specific the developed method is. The forced degradation studies have been carried out to observe the specificity of the developed HPLC method for the determination of VZ. The solution of VZ has been prepared and exposed to the particular stress conditions for 2 hours and afterward subjected to the HPLC analysis. The results obtained are given in Table 8. It has been found that maximum degradation of VZ occurs via alkaline hydrolysis (82.33%), followed by UV light (10.64%) and heat (7.55%). The maximum degradation of VZ in alkaline hydrolysis is due to the higher

**Table 4. Calibration data for the determination of Voriconazole by HPLC.**

| $\lambda_{max}$ (nm) | 254 |
|---|---|
| **Linearity** | |
| Range (µg/100 mL) | 3.5–45.0 |
| Correlation coefficient | 0.9999 |
| Slope | 212571 |
| SE of slope | 36376 |
| Intercept | 13952 |
| SE of intercept | 25901 |
| SD of intercept | 68529 |
| Accuracy (%) ± SD | 99.96–101.25±0.56 |
| Precision (%RSD) | 0.56 |
| LOD (µg/100 ml) | 1.06 |
| LOQ (µg/100 ml) | 3.22 |

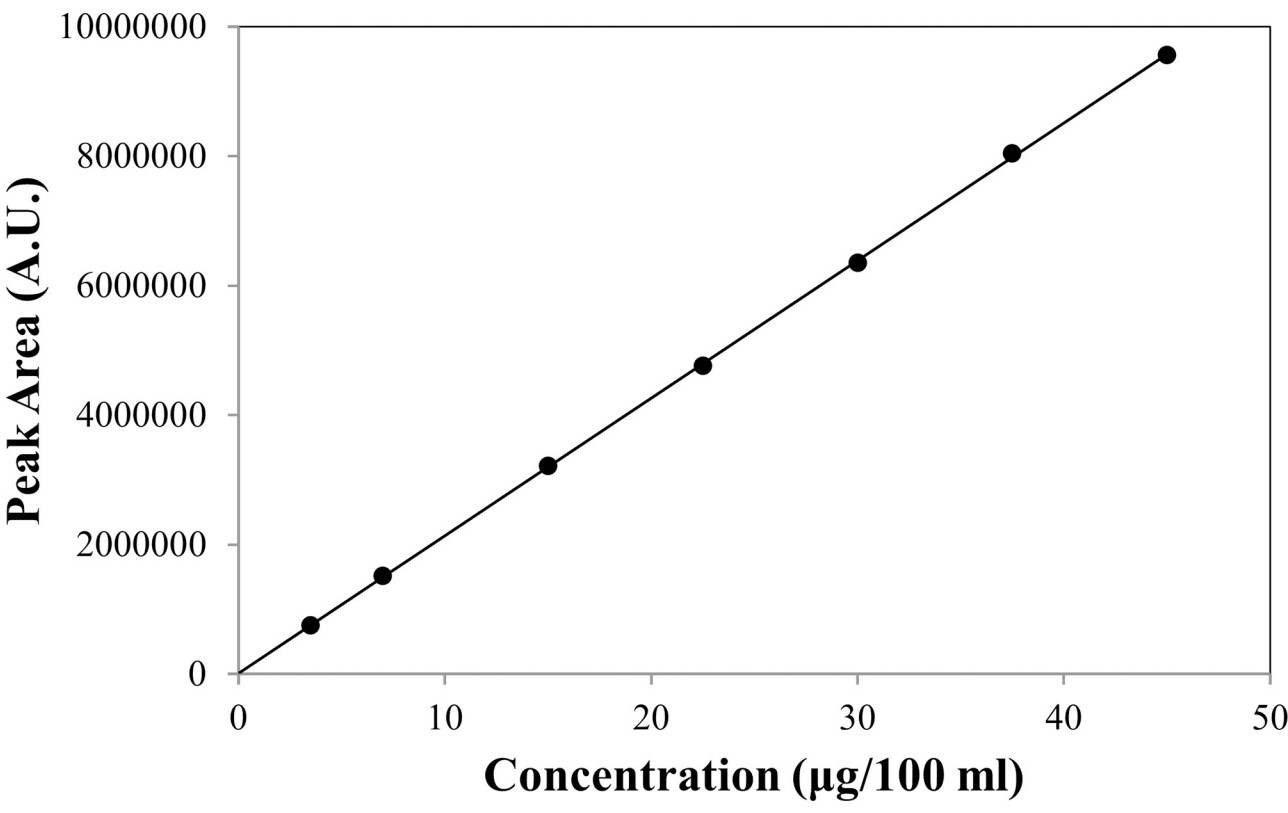

**Fig 3. Calibration curve for VZ in the concentration range of 3.5–45.0 µg/100 ml.**

susceptibility of its anionic form to alkaline conditions, as reported earlier [83]. However, minimum degradation has been found in the case of acid hydrolysis (3.71%) and oxidative stress (2.55%) [83]. This may be due to the lower sensitivity of VZ in its cationic form to acid hydrolysis. Acid, base hydrolysis, and oxidation degradation show no interfering peak at the analyte's retention time and no degradation peak because of using a UV detector instead of a PDA detector, which is not available in our laboratory.

**Table 5. Accuracy of the proposed HPLC method.**

| Added Concentration (µg/100 mL) | Found Concentration (µg/100 mL) | Recovery (%) | Mean recovery (%) ± SD (% RSD) |
|---|---|---|---|
| 15.0 | 14.99 | 99.93 | |
| | 15.05 | 100.33 | 99.76±0.68 (0.68) |
| | 14.85 | 99.00 | |
| 30.0 | 30.11 | 100.37 | |
| | 29.89 | 99.63 | 100.0±0.37 (0.37) |
| | 30.01 | 100.03 | |
| 45.0 | 45.11 | 100.24 | |
| | 44.96 | 99.91 | 100.1±0.18 (0.18) |
| | 45.09 | 100.20 | |

**Table 6. The precision of the proposed HPLC method.**

| Precision | Added Concentration (µg/100 mL) | Found Concentration (µg/100 mL) | Recovery (%) | Mean recovery (%) ± SD (% RSD) |
|---|---|---|---|---|
| Repeatability | 30.0 | 29.89 | 99.63 | |
| | | 30.11 | 100.3 | |
| | | 30.03 | 100.1 | |
| | | 29.88 | 99.60 | 99.98±0.31 (0.31) |
| | | 30.01 | 100.0 | |
| | | 30.11 | 100.3 | |
| Intermediate | 30.0 | 30.11 | 100.3 | |
| | | 30.05 | 100.1 | |
| | | 29.87 | 99.57 | 100.0±0.32 (0.32) |
| | | 30.11 | 100.3 | |
| | | 29.95 | 99.83 | |
| | | 30.14 | 100.4 | |

**Table 7. Robustness of the proposed HPLC method.**

| Robustness parameters | Accuracy (%) ± SD | Precision (% RSD) | Student *t*-test[a] |
|---|---|---|---|
| Wavelength (±2nm) | | | |
| 252 | 100.01±0.11 | 0.11 | 0.68 |
| 256 | 99.86±0.22 | 0.22 | 0.45 |
| Temperature (±5˚C) | | | |
| 25 | 98.99±0.88 | 0.88 | 0.15 |
| 35 | 100.1±0.32 | 0.32 | 0.74 |
| Flow rate (mL/min) | | | |
| 0.8 | 100.1±0.44 | 0.44 | 0.11 |
| 1.2 | 99.99±0.51 | 0.51 | 0.48 |

[a]At 95% confidence interval the two-degree freedom is 4.303, whereas the calculated values are in the range of 0.11–0.74 which is less than the tabulated values indicating that there is no difference between the proposed method conditions and deliberate changes

## Application of proposed HPLC method

The developed and validated HPLC method is found to be accurate (Table 5), precise (Table 6), less time-consuming ($t_R$ = 3.782, Table 3), simple, sensitive (Table 4), and economic (less amount of organic solvent (acetonitrile) used) as compared to that of the previously

**Table 8. Specificity data for the determination of VZ through the proposed HPLC method.**

| Conditions | Recovery (%) | Degradation (%) |
|---|---|---|
| Acid Hydrolysis (0.1 N HCl) | 97.19 | 3.71 |
| Base Hydrolysis (0.1 N NaOH) | 17.67 | 82.33 |
| Oxidation (3% $H_2O_2$) | 97.55 | 2.55 |
| UV light | 89.36 | 10.64 |
| Temperature (100˚C) | 92.55 | 7.55 |

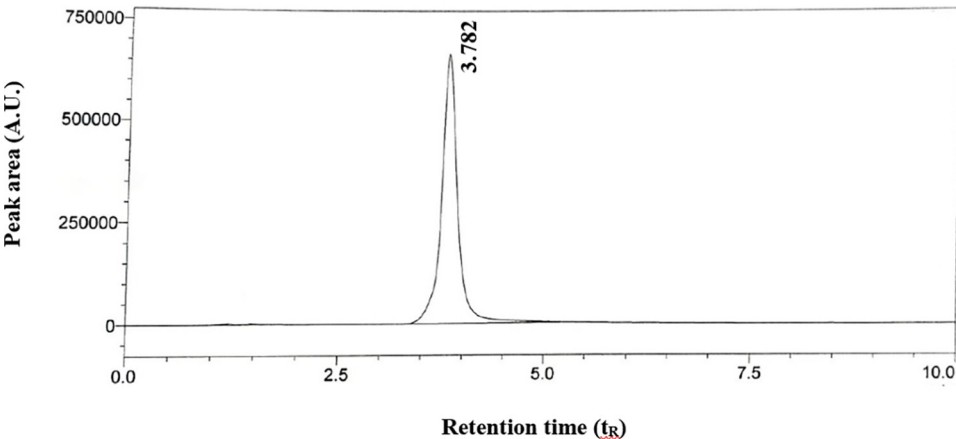

**Fig 4. Chromatogram of voriconazole in gel formulation using the proposed HPLC method.**

reported HPLC methods. So, therefore, the reliability, sensitivity, and selectivity of the proposed HPLC method have been checked by applying it to the gel formulations. The proposed HPLC method was applied to determine the concentration of VZ in the gel formulation. The gel formulation was dissolved in ethanol and subjected to the HPLC analysis. The obtained chromatogram is given in Fig 4. It has been found that the excipients (i.e carbomer, PG, TEA) used for the preparation of the gel do not show interference. The results obtained were found to be in the range of 98.00–102.7% with a %RSD of 0.88–1.31 (Table 9), so, therefore the proposed method could be commercially used for the analysis of VZ in gel formulations.

## Conclusion

The HPLC method has been developed and validated for the assay of voriconazole (VZ) in pure form and its gel formulation. The proposed method is rapid, simple, sensitive, accurate, precise, robust, and specific for the estimation of VZ. This HPLC method is stability-indicating because of its specificity in determining VZ in the presence of degradation products. Also, this method is less time-consuming as the VZ peak appears at 3.8 min and is economical due to the simple mobile phase system with lesser quantity. The developed HPLC method is also successfully applied to the gel formulations for the estimation of VZ.

**Table 9. Assay of VZ in gel formulations using the proposed HPLC method.**

| Added (g/100gm) | Found | Recovery (%) | Mean Recovery (%)±SD (%RSD) |
|---|---|---|---|
| 0.50 | 0.50 | 100.0 | |
| | 0.51 | 100.2 | 99.4±1.21 (1.22) |
| | 0.49 | 98.00 | |
| 1.50 | 1.52 | 101.3 | |
| | 1.50 | 100.0 | 101.3±1.33 (1.31) |
| | 1.54 | 102.7 | |
| 3.00 | 3.05 | 101.7 | |
| | 3.01 | 100.3 | 100.7±0.88 (0.88) |
| | 3.00 | 100.0 | |

## Author Contributions

**Conceptualization:** Samina Sheikh.

**Data curation:** Samina Sheikh, Sadaf Ibrahim, Ambreen Huma, Aisha Jabeen, Zubair Anwar.

**Formal analysis:** Samina Sheikh.

**Investigation:** Samina Sheikh, Sadaf Ibrahim, Ambreen Huma, Aisha Jabeen, Zubair Anwar.

**Supervision:** Anwar Ejaz Beg, Mirza Tasawer Baig.

**Writing – original draft:** Samina Sheikh.

**Writing – review & editing:** Zubair Anwar.

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
