## [Decision Letter · Decision Letter 0]

5 Sep 2024

PONE-D-24-28984High Performance Liquid Chromatographic Method for Determination of Voriconazole in Pure and Gel FormulationPLOS ONE

Dear Dr. Anwar,

Thank you for submitting your manuscript to PLOS ONE. After careful consideration, we feel that it has merit but does not fully meet PLOS ONE’s publication criteria as it currently stands. Therefore, we invite you to submit a revised version of the manuscript that addresses the points raised during the review process.

We look forward to receiving your revised manuscript.

Kind regards,

António Machado

Academic Editor

PLOS ONE Journal Requirements: 

3. Please upload a new copy of Figures 1 to 4 as the detail is not clear. Please follow the link for more information:

https://blogs.plos.org/plos/2019/06/looking-good-tips-for-creating-your-plos-figures-graphics/
https://blogs.plos.org/plos/2019/06/looking-good-tips-for-creating-your-plos-figures-graphics/

**Additional Editor Comments:**

I am pleased to say that we are excited about the possibility of publishing your work. However, reviewers reported several concerns, mixed decisions between major revisions and rejection, and the need for major revisions in the original version of the manuscript. Please read carefully all reviewers’ reports addressing and answering all comments and suggestions. So, I kindly invite the authors to realize a thoughtful revision of the submitted manuscript to achieve publication endorsement by most reviewers. Thank you and best regards,

António Machado

Reviewers' comments:

Reviewer's Responses to Questions

**Comments to the Author**

1. Is the manuscript technically sound, and do the data support the conclusions?

Reviewer #1: Yes

Reviewer #2: No

Reviewer #3: Yes

Reviewer #4: No

Reviewer #5: No

Reviewer #6: Partly

2. Has the statistical analysis been performed appropriately and rigorously? 

Reviewer #1: Yes

Reviewer #2: No

Reviewer #3: Yes

Reviewer #4: No

Reviewer #5: No

Reviewer #6: No

3. Have the authors made all data underlying the findings in their manuscript fully available?

Reviewer #1: Yes

Reviewer #2: Yes

Reviewer #3: Yes

Reviewer #4: No

Reviewer #5: Yes

Reviewer #6: Yes

4. Is the manuscript presented in an intelligible fashion and written in standard English?

Reviewer #1: Yes

Reviewer #2: No

Reviewer #3: Yes

Reviewer #4: No

Reviewer #5: No

Reviewer #6: No

5. Review Comments to the Author

Reviewer #1: Following points should be confirmed again;

1. The figure number "1a" is repeated... correct the same

2. Fig 2 shows retention time 3.782 however in the text it is mentioned as 4.14 against the same figure

3. Check the unit in figure 3 ....microgram per 100 gm or ml

4. Table 2 data for theoretical plates and tailing factor and its description in the text is not matching

Reviewer #2: The current manuscript mainly focuses on determination of voriconazole in pure and gel formulation which has been already reported in various study as also mentioned in introduction section. I feel there is novelty issue.

Except that I have some observation as below:

The introduction is not well-written.

Purity of Voriconazole should be mentioned.

Preparation of Gel formulation must be supported by suitable reference.

The quality of the English language throughout the manuscript is inferior (for ex; see the section Chromatographic Condition).

The extract concentration of standard used for calibration curve preparation should be mentioned in manuscript.

The presentation of the results and discussion is not well organized.

Figure 2 & 4 looks similar.

Chemical structure should be self-drawn.

Some references are not as per journal guidelines.

Reviewer #3: The manuscript is good and clear however some comments below should be considered in the revised version.

-No literature was provided for the HPLC and its application or validation parameters selection, please use the following reference list:

#Yehia, A.M. and Mohamed, H.M., 2016. Journal of Separation Science, 39(11), pp.2114-2122.

#Yehia, A.M.,et al., 2017. Chromatographia, 80, pp.99-107.

#Yehia, A.M., et al, 2018. Separation Science Plus, 1(6), pp.395-403.

#Weshahy, S. et al, 2020. Microchemical Journal, 157, p.105047.

#Yehia, A.M., et al, 2021. Chromatographia, 84, pp.1-11.

#Yehia, A.M. and Essam, H.M., 2016. Journal of Separation Science, 39(17), pp.3357-3367.

-The final volume should be specified in the preparation of solution part.

-Technically, Liter should be abbreviated in L in ml.

-In LOD and LOQ calculation "σ" is the standard deviation of what?

-Please specify the limits in varying the flow rate (??ml/min)

-Robustness should be calculated collectively for one parameter change but not for each level within the same parameter.

-Please use reliable figure especially for the chemical structure.

Reviewer #4: PONE-D-24-28984

Authors reported High-Performance Liquid Chromatographic Method for Determination of Voriconazole in Pure and Gel Formulation, as a reviewer, I do not question the importance of the proposed work. However, a significant number of analytical methods are already published. The development presented here is not a genuinely innovative contribution and does not meet the journal's standards. It seems that the quantification of the analyte from a simple matrix and the development of new HPLC methods without specific challenges considering the principles of separation science is a contribution of little relevance to the journal's audience.

Few reported methods:

1. G. Srinubabu, Ch. A.I. Raju, N. Sarath, P. Kiran Kumar, J.V.L.N. Seshagiri Rao, Development and validation of a HPLC method for the determination of voriconazole in pharmaceutical formulation using an experimental design, Talanta, Volume 71, Issue 3, 2007, Pages 1424-1429, https://doi.org/10.1016/j.talanta.2006.04.042.

2. Khetre AB, Sinha PK, Damle MC, Mehendre R. Development and Validation of Stability Indicating RP-HPLC Method for Voriconazole. Indian J Pharm Sci. 2009 Sep;71(5):509-14. doi: 10.4103/0250-474X.58178. PMID: 20502568; PMCID: PMC2866341.

3. Sahar Yousefian, Farzaneh Dastan, Majid Marjani, Payam Tabarsi, Saghar Barati, Nahid Shahsavari, Farzad Kobarfard. Determination of Voriconazole Plasma Concentration by HPLC Technique and Evaluating Its Association with Clinical Outcome and Adverse Effects in Patients with Invasive Aspergillosis https://doi.org/10.1155/2021/5497427

4. Yasu, T.; Nomura, Y.; Gando, Y.; Matsumoto, Y.; Sugita, T.; Kosugi, N.; Kobayashi, M. High-Performance Liquid Chromatography for Ultra-Simple Determination of Plasma Voriconazole Concentration. J. Fungi 2022, 8, 1035. https://doi.org/10.3390/jof8101035

5. Zhang M, Moore GA, Barclay ML, Begg EJ. 2013. A Simple High-Performance Liquid Chromatography Method for Simultaneous Determination of Three Triazole Antifungals in Human Plasma. Antimicrob Agents Chemother 57:. https://doi.org/10.1128/aac.00768-12

Then, based on the above observations, my overall recommendation is to reject the manuscript in its present form.

Reviewer #5: I strongly recommend rejecting the manuscript considering missing novelty in the manuscript. There are so many reported methods available on determination of Voriconazole. Few of them are mentioned below for your reference:

https://www.ncbi.nlm.nih.gov/pmc/articles/PMC2866341/

https://dergipark.org.tr/tr/download/article-file/166318

https://www.sciencedirect.com/science/article/abs/pii/S0039914007000021

Reviewer #6: The research paper details the development and validation of an RP-HPLC method for determining Voriconazole (VZ) in both pure and gel formulations. The method is described as quick, accurate, and robust, with good precision and a short retention time. However, the novelty of the work is questionable. RP-HPLC is a well-established technique for the analysis of Voriconazole, as evidenced by the numerous previous studies referenced in the introduction. The paper does not clearly distinguish how this method significantly improves upon or differs from existing methods. However, without clear evidence or comparison showing that this method significantly overcomes these limitations, the rationale lacks full support. The application to gel formulations is a reasonable extension, though this aspect alone does not establish strong novelty. Although the short retention time and the method’s application to gel formulations are positive features, these aspects alone do not constitute substantial innovation, especially without a direct comparison to other methods in terms of time, cost, or sensitivity.

Strengths:

1. Comprehensive Validation: The method is thoroughly validated according to ICH guidelines, covering system suitability, linearity, accuracy, precision, specificity, and robustness.

2. Short Retention Time: A retention time of less than 4 minutes is efficient for routine analysis.

3. Application to Gel Formulations: The method’s successful application to both pure and gel formulations broadens its utility.

Areas for Improvement:

1. Comparative Analysis for Novelty: To better demonstrate novelty, the paper should include a comparison with existing RP-HPLC methods, particularly highlighting specific improvements in terms of time, cost, simplicity, or sensitivity.

2. Detailed Method Development Justification: The paper could benefit from a more detailed explanation of the choice of mobile phase composition, flow rate, and other chromatographic conditions, perhaps by comparing these with those used in similar studies.

3. Robustness Testing: While the method's robustness is addressed, the paper should detail the exact parameters tested and their impact on the analysis, along with statistical data to support the findings.

4. Discussion on Specificity and Degradation Studies: The discussion of degradation studies and the method's specificity could be enhanced with more detailed results and interpretations, particularly in the context of forced degradation.

6. PLOS authors have the option to publish the peer review history of their article (what does this mean?). If published, this will include your full peer review and any attached files.

Reviewer #1: **Yes: **Dinesh Rishipathak

Reviewer #2: No

Reviewer #3: No

Reviewer #4: No

Reviewer #5: No

Reviewer #6: No

---

## [Author Response · Author response to Decision Letter 0]

20 Nov 2024

REPLIES TO REVIEWER'S COMMENTS

Reviewer 1

1. The figure number "1a" is repeated... correct the same.

It has now been corrected.

2. Fig 2 shows retention time 3.782 however in the text it is mentioned as 4.14 against the same figure.

It has now been corrected

3. Check the unit in figure 3 ....microgram per 100 gm or ml.

It is microgram/100 ml.

4. Table 2 data for theoretical plates and tailing factor and its description in the text is not matching.

It has now been corrected.

Reviewer 2

The current manuscript mainly focuses on the determination of voriconazole in pure and gel formulation which has been already reported in various study as also mentioned in introduction section. I feel there is novelty issue.

The novelty of the present work lies in its efficient and streamlined approach to the analysis of voriconazole in both pure form and gel formulations. Unlike traditional methods, this technique is significantly less time-consuming and requires a minimal number and amount of solvents, making it more environmentally friendly and cost-effective. Furthermore, it eliminates the need for any sample pretreatment, simplifies the analytical process, and reduces the risk of potential errors associated with sample handling. This innovative method offers a rapid, straightforward, and resource-efficient alternative for the accurate quantification of voriconazole.

Except that I have some observation as below:

The introduction is not well-written.

The introduction section has now been improved.

The purity of Voriconazole should be mentioned.

It has now been mentioned in the materials and methods section.

Preparation of Gel formulation must be supported by suitable reference.

Suitable references have now been added to prepare the gel section.

The quality of the English language throughout the manuscript is inferior (for ex; see the section Chromatographic Condition).

The quality of the language throughout the manuscript has now been improved by using the licensed version of Grammarly.

The extract concentration of the standard used for calibration curve preparation should be mentioned in manuscript.

It has now been mentioned in the preparation of the reference standard solution section

The presentation of the results and discussion is not well organized.

The presentation of results and discussion has now been improved.

Figure 2 & 4 looks similar.

They have now been changed.

Chemical structure should be self-drawn.

The chemical structures have now been improved and self drawn.

Some references are not as per journal guidelines.

They have now been corrected.

Reviewer 3

The manuscript is good and clear however some comments below should be considered in the revised version.

No literature was provided for the HPLC and its application or validation parameters selection, please use the following reference list:

#Yehia, A.M. and Mohamed, H.M., 2016. Journal of Separation Science, 39(11), pp.2114-2122.

#Yehia, A.M.,et al., 2017. Chromatographia, 80, pp.99-107.

#Yehia, A.M., et al, 2018. Separation Science Plus, 1(6), pp.395-403.

#Weshahy, S. et al, 2020. Microchemical Journal, 157, p.105047.

#Yehia, A.M., et al, 2021. Chromatographia, 84, pp.1-11.

#Yehia, A.M. and Essam, H.M., 2016. Journal of Separation Science, 39(17), pp.3357-3367.

The literature on the HPLC and its application previously used has now been given in the introduction section along with Table 1 possessing all the necessary information about the previously developed HPLC methods. The validation of the proposed method has been carried out using the guidelines of the International Council on Harmonization (2005) and has been mentioned in the validation section.

-The final volume should be specified in the preparation of the solution part.

It has now been mentioned in the solution part.

Technically, Liter should be abbreviated in L in ml.

It has now been corrected in the whole manuscript.

In LOD and LOQ calculation "σ" is the standard deviation of what?

It is the standard deviation of the intercept, and it is mentioned under the formula given in the LOD and LOQ section.

Please specify the limits in varying the flow rate (??ml/min)

It has now been mentioned in the robustness section.

Robustness should be calculated collectively for one parameter change but not for each level within the same parameter.

The robustness of the proposed HPLC method for the determination of voriconazole has been evaluated using the guidelines of the International Council on Harmonization (2005). We have made deliberate changes in wavelength (±2 nm), temperature (±5 oC), and flow rate (±0.2 mL/min) of the mobile phase. After making these deliberate changes we have determined the accuracy, precision, and overall performance of the proposed HPLC method for the determination of voriconazole. 

Please use reliable figure especially for the chemical structure.

They have now been improved.

Reviewer 4

Authors reported High-Performance Liquid Chromatographic Method for Determination of Voriconazole in Pure and Gel Formulation, as a reviewer, I do not question the importance of the proposed work. However, a significant number of analytical methods are already published. The development presented here is not a genuinely innovative contribution and does not meet the journal's standards. It seems that the quantification of the analyte from a simple matrix and the development of new HPLC methods without specific challenges considering the principles of separation science is a contribution of little relevance to the journal's audience.

Few reported methods:

1. G. Srinubabu, Ch. A.I. Raju, N. Sarath, P. Kiran Kumar, J.V.L.N. Seshagiri Rao, Development and validation of a HPLC method for the determination of voriconazole in pharmaceutical formulation using an experimental design, Talanta, Volume 71, Issue 3, 2007, Pages 1424-1429, https://doi.org/10.1016/j.talanta.2006.04.042.

2. Khetre AB, Sinha PK, Damle MC, Mehendre R. Development and Validation of Stability Indicating RP-HPLC Method for Voriconazole. Indian J Pharm Sci. 2009 Sep;71(5):509-14. doi: 10.4103/0250-474X.58178. PMID: 20502568; PMCID: PMC2866341.

3. Sahar Yousefian, Farzaneh Dastan, Majid Marjani, Payam Tabarsi, Saghar Barati, Nahid Shahsavari, Farzad Kobarfard. Determination of Voriconazole Plasma Concentration by HPLC Technique and Evaluating Its Association with Clinical Outcome and Adverse Effects in Patients with Invasive Aspergillosis https://doi.org/10.1155/2021/5497427

4. Yasu, T.; Nomura, Y.; Gando, Y.; Matsumoto, Y.; Sugita, T.; Kosugi, N.; Kobayashi, M. High-Performance Liquid Chromatography for Ultra-Simple Determination of Plasma Voriconazole Concentration. J. Fungi 2022, 8, 1035. https://doi.org/10.3390/jof8101035

5. Zhang M, Moore GA, Barclay ML, Begg EJ. 2013. A Simple High-Performance Liquid Chromatography Method for Simultaneous Determination of Three Triazole Antifungals in Human Plasma. Antimicrob Agents Chemother 57:. https://doi.org/10.1128/aac.00768-12.

The novelty of the present work lies in its efficient and streamlined approach to the analysis of voriconazole in both pure form and gel formulations. Unlike traditional methods, this technique is significantly less time-consuming and requires a minimal number and amount of solvents, making it more environmentally friendly and cost-effective. Furthermore, it eliminates the need for any pretreatment of the sample, simplifying the analytical process and reducing the risk of potential errors associated with sample handling. This innovative method offers a rapid, straightforward, and resource-efficient alternative for the accurate quantification of voriconazole.

Then, based on the above observations, my overall recommendation is to reject the manuscript in its present form.

Reviewer 5

I strongly recommend rejecting the manuscript considering missing novelty in the manuscript. There are so many reported methods available on determination of Voriconazole. Few of them are mentioned below for your reference:

https://www.ncbi.nlm.nih.gov/pmc/articles/PMC2866341/

https://dergipark.org.tr/tr/download/article-file/166318

https://www.sciencedirect.com/science/article/abs/pii/S0039914007000021

The novelty of the present work lies in its efficient and streamlined approach to the analysis of voriconazole in both pure form and gel formulations. Unlike traditional methods, this technique is significantly less time-consuming and requires minimal solvents, making it more environmentally friendly and cost-effective. Furthermore, it eliminates the need for any sample pretreatment, simplifying the analytical process and reducing the risk of potential errors associated with sample handling. This innovative method offers a rapid, straightforward, and resource-efficient alternative for the accurate quantification of voriconazole.

Reviewer 6

The research paper details the development and validation of an RP-HPLC method for determining Voriconazole (VZ) in both pure and gel formulations. The method is described as quick, accurate, and robust, with good precision and a short retention time. However, the novelty of the work is questionable. RP-HPLC is a well-established technique for the analysis of Voriconazole, as evidenced by the numerous previous studies referenced in the introduction. The paper does not clearly distinguish how this method significantly improves upon or differs from existing methods. However, without clear evidence or comparison showing that this method significantly overcomes these limitations, the rationale lacks full support. The application to gel formulations is a reasonable extension, though this aspect alone does not establish strong novelty. Although the short retention time and the method’s application to gel formulations are positive features, these aspects alone do not constitute substantial innovation, especially without a direct comparison to other methods in terms of time, cost, or sensitivity.

Strengths:

The novelty of the present work lies in its efficient and streamlined approach to the analysis of voriconazole in both pure form and gel formulations. Unlike traditional methods, this technique is significantly less time-consuming and requires a minimal number and number of solvents, making it more environmentally friendly and cost-effective. Furthermore, it eliminates the need for any pretreatment of the sample, simplifies the analytical process, and reduces the risk of potential errors associated with sample handling. This innovative method offers a rapid, straightforward, and resource-efficient alternative for the accurate quantification of voriconazole.

1. Comprehensive Validation: The method is thoroughly validated according to ICH guidelines, covering system suitability, linearity, accuracy, precision, specificity, and robustness.

2. Short Retention Time: A retention time of less than 4 minutes is efficient for routine analysis.

3. Application to Gel Formulations: The method’s successful application to both pure and gel formulations broadens its utility.

Areas for Improvement:

1. Comparative Analysis for Novelty: To better demonstrate novelty, the paper should include a comparison with existing RP-HPLC methods, particularly highlighting specific improvements in terms of time, cost, simplicity, or sensitivity.

The comparative analysis of the novelty of the proposed HPLC method has been included in the manuscript by the addition of Table 1 containing details of the previously developed HPLC methods for the determination of voriconazole. The following lines have been added in the introduction section.

The details of the previously developed HPLC methods for the determination of VZ in pure and pharmaceutical formulations are given in Table 1. Previous HPLC methods developed for the determination of VZ are time-consuming, tedious, expensive and need pretreatment of the samples. So, therefore, the present study aims to develop and validate an easy, affordable, sensitive, reliable, precise and robust HPLC method for the determination of VZ in pure and gel formulations.

Also, the improvements in the proposed HPLC method are highlighted in the application of the method section by the addition of the following lines.

The developed and validated HPLC method is found to be accurate (Table 5), precise (Table 6), less time-consuming (tR = 3.782, Table 3), simple, sensitive (Table 4), and economic (less amount of organic solvent (acetonitrile) used) as compared to that of the previously reported HPLC methods. So, therefore, the reliability, sensitivity, and selectivity of the proposed HPLC method have been checked by applying it to the gel formulations. 

2. Detailed Method Development Justification: The paper could benefit from a more detailed explanation of the choice of mobile phase composition, flow rate, and other chromatographic conditions, perhaps by comparing these with those used in similar studies.

The justification of the development and validation of proposed HPLC method for the determination of VZ has now been given by the addition of the following lines in the results and discussion section.

The developed and validated HPLC method improves on previously reported HPLC methods by addressing the demands of sensitivity, selectivity, accuracy, precision, robustness, and specificity for the quantification of VZ. Various parameters have been studied to optimize the analysis conditions, including solvent system composition (acetonitrile:water, 10:90, 15:85, 20:80, 25:75, 30:70, 40:60, and 50:50, v/v), pH of the mobile phase (2.0, 2.5, 3.0, 3.5, 4.0, 4.5, 5.0, and 6.5), and flow rate of the mobile phase (0.1, 0.2, 0.3, 0.4, 0.6, 0.8, 1.0 mL/min). After multiple trials using the above-mentioned conditions, the proposed method has been validated using acetonitrile:water (50:50, v/v) with a flow rate of 1.0 mL/min at pH 5.0 to obtain accurate, precise, robust, reliable, and specific results for the estimation of VZ. 

3. Robustness Testing: While the method's robustness is addressed, the paper should detail the exact parameters tested and their impact on the analysis, along with statistical data to support the findings.

The exact parameters tested have now been mentioned in the materials and methods and results and discussions sections. Also, the outcomes of these changes have been given in Table 7 in terms of the accuracy and precision of the proposed method that the proposed method is reliable, accurate and precise. The statistical comparison between the original conditions of the proposed method and after making deliberate changes has been carried out by applying the student t-test at 95% confidence interval. It has been found that the calculated t values are less than that of tabulated values indicating that there is no significant difference between the results obtained after making deliberate changes and the original conditions of the proposed method. The following lines have been added in the robustness section.

It demonstrates that the testing method is unaffected by any little intentional alterations and demonstrates the method's compatibility and durability. The deliberate changes in temperature (±5 oC), wavelength (±2 nm) and flow rate (± 0.2 mL/min), as well as the accuracy and precision data collected following those adjustments, are given in Table 7. The accuracy and precision (%RSDs) for the identification of VZ after deliberate changes are 99.86-100.01,98.99-100.1, 99.99-100.1 and 0.11-0.22, 0.32-0.88, 0.44-0.51, respectively. The statistical comparison between the proposed method conditions and after making the deliberate changes has been carried out (Table 7). It has been found that there is no significant difference between the proposed method conditions and the deliberate changes made. The calculated t values are less than that of tabulated t values indicating that there is no significant difference and confirming that the method is reliable, sensitive, accurate, and precise. 

4. Discussion on Specificity and Degradation Studies: The discussion of degradation studies and the method's specificity could be enhanced with more detailed results and interpretations, particularly in the context of forced degradation.

The specificity and degradation section has now been improved and tried to explain in more detai

---

## [Decision Letter · Decision Letter 1]

2 Dec 2024

High Performance Liquid Chromatographic Method for Determination of Voriconazole in Pure and Gel Formulation

PONE-D-24-28984R1

Dear authors,

I am pleased to inform you that both reviewers enjoyed the manuscript very much and endorsed the revised manuscript for publication. Please try to briefly enhance the novelty of the work during the process editorial process as recommended by Reviewer 6.

Thank you for choosing Plos ONE journal to publish your study.

Best regards,

António Machado

Reviewers' comments:

Reviewer's Responses to Questions

**Comments to the Author**

1. If the authors have adequately addressed your comments raised in a previous round of review and you feel that this manuscript is now acceptable for publication, you may indicate that here to bypass the “Comments to the Author” section, enter your conflict of interest statement in the “Confidential to Editor” section, and submit your "Accept" recommendation.

Reviewer #1: All comments have been addressed

Reviewer #3: All comments have been addressed

Reviewer #6: All comments have been addressed

2. Is the manuscript technically sound, and do the data support the conclusions?

Reviewer #1: Yes

Reviewer #3: Yes

Reviewer #6: Partly

3. Has the statistical analysis been performed appropriately and rigorously? 

Reviewer #1: Yes

Reviewer #3: Yes

Reviewer #6: No

4. Have the authors made all data underlying the findings in their manuscript fully available?

Reviewer #1: Yes

Reviewer #3: Yes

Reviewer #6: Yes

5. Is the manuscript presented in an intelligible fashion and written in standard English?

Reviewer #1: Yes

Reviewer #3: Yes

Reviewer #6: Yes

6. Review Comments to the Author

Reviewer #1: (No Response)

Reviewer #3: All comments have been addressed in the revised version and the work is now suitable for publication. No conflict of dual publication or research ethics.

Reviewer #6: What is Novelty of your published method in terms of sensitivity, mobile phase, retention time etc.

7. PLOS authors have the option to publish the peer review history of their article (what does this mean?). If published, this will include your full peer review and any attached files.

Reviewer #1: No

Reviewer #3: No

Reviewer #6: No

---

## [Editor Report · Acceptance letter]

4 Dec 2024

PONE-D-24-28984R1 

PLOS ONE

Dear Dr. Anwar, 

I'm pleased to inform you that your manuscript has been deemed suitable for publication in PLOS ONE. Congratulations! Your manuscript is now being handed over to our production team.

Kind regards, 

on behalf of

Dr. António Machado 

Academic Editor

PLOS ONE